



# RainNet v1.0: a convolutional neural network for radar-based precipitation nowcasting

Georgy Ayzel[1], Tobias Scheffer[2], and Maik Heistermann[1]

[1]Institute for Environmental Sciences and Geography, University of Potsdam, Potsdam, Germany
[2]Department of Computer Science, University of Potsdam, Potsdam, Germany

**Correspondence:** Georgy Ayzel (ayzel@uni-potsdam.de)

**Abstract.** In this study, we present RainNet, a deep convolutional neural network for radar-based precipitation nowcasting. Its design was inspired by the U-Net and SegNet families of deep learning models which were originally designed for binary segmentation tasks. RainNet was trained to predict continuous precipitation intensities at a lead time of five minutes, using several years of quality-controlled weather radar composites provided by the German Weather Service (DWD). That data
set covers Germany with a spatial domain of 900×900 km, and has a resolution of 1 km in space and 5 minutes in time. Independent verification experiments were carried out on eleven summer precipitation events from 2016 to 2017. In order to achieve a lead time of one hour, a recursive approach was implemented by using RainNet predictions at five minutes lead time as model input for longer lead times. In the verification experiments, trivial Eulerian persistence and a conventional model based on optical flow served as benchmarks. The latter is available in the *rainymotion* library, and had previously been shown
to outperform DWD's operational nowcasting model for the same set of verification events.

RainNet significantly outperforms the benchmark models at all lead times up to 60 minutes for the routine verification metrics Mean Absolute Error (MAE) and Critical Success Index (CSI, at intensity thresholds of 0.125, 1, and 5 mm h$^{-1}$). Apart from its superiority in terms of MAE and CSI, an undesirable property of RainNet predictions is, however, the level of spatial smoothing. At a lead time of five minutes, an analysis of Power Spectral Density confirmed a significant loss of spectral
power at length scales of 16 km and below. Obviously, RainNet had learned an optimal level of smoothing to produce a nowcast at 5 minutes lead time. In that sense, the loss of spectral power at small scales is informative, too, as it reflects the limits of predictability as a function of spatial scale. Beyond the lead time of five minutes, however, the increasing level of smoothing is a mere artifact – an analogue to numerical diffusion – that is not a property of RainNet itself, but of its recursive application. In the context of early warning, the smoothing is particularly unfavourable since pronounced features of intense precipitation tend
to get lost over longer lead times. Hence, we propose several options to address this issue in prospective research, including an adjustment of the loss function for model training, model training for longer lead times, and the prediction of threshold exceedance in terms of a binary segmentation task. Furthermore, we suggest additional input data that could help to better identify situations with imminent precipitation dynamics. The model code, pretrained weights, and training data are provided in open repositories as an input to such future studies.



# 1 Introduction


The term *nowcasting* refers to forecasts of precipitation field movement and evolution at high spatiotemporal resolution (1–10 minutes, 100–1000 meters) and short lead times (minutes to a few hours). Nowcasts have become popular not only to a broad civil community for planning everyday activities; they are particularly relevant as part of early warning systems for heavy rainfall, and related impacts such as flash floods or landslides. While the recent advances in high-performance computing

and data assimilation significantly improved numerical weather prediction (NWP) (Bauer et al., 2015), the computational resources required to forecast precipitation field dynamics at very high spatial and temporal resolution are typically prohibitive for the frequent update cycles (5–10 minutes) that are required for operational nowcasting systems. Furthermore, the heuristic extrapolation of precipitation dynamics that are observed by weather radars still outperform NWP forecasts at short lead times (Lin et al., 2005; Sun et al., 2014). Thus, the development of new nowcasting systems based on parsimonious, but reliable and

fast techniques, remains an essential trait in both atmospheric and natural hazards research.

There are many nowcasting systems which work operationally all around the world to provide precipitation nowcasts (Reyniers, 2008; Wilson et al., 1998). These systems, in their core, utilize a two-step procedure that was originally suggested by Austin and Bellon (1974), consisting of tracking and extrapolation. In the tracking step, a velocity is obtained from a series of consecutive radar images. In the extrapolation step, that velocity is used to propagate the most recent precipitation observation

into the future. Various flavors and variations of this fundamental idea have been developed and operationalized over the past decades, and provide value to users of corresponding products. Still, the fundamental approach to nowcasting has not changed much over the recent years – a situation that might change with the increasing popularity of deep learning in various scientific disciplines.

*Deep learning* refers to machine-learning methods for artificial neural networks with "deep" architectures. Rather than

relying on engineered features, deep learning derives low-level image features on the lowest layers of a hierarchical network, and increasingly abstract features on the high-level network layers, as part of the solution of an optimization problem based on training data (LeCun et al., 2015). Deep learning took its rise from the field of computer science when it started to dramatically outperform reference methods in image classification (Krizhevsky et al., 2012), machine translation (Sutskever et al., 2014), followed by speech recognition (LeCun et al., 2015). Three main reasons caused this substantial breakthrough in predictive

efficacy: the availability of "big data" for model training, the development of activation functions and network architectures that result in numerically stable gradients across many network layers (Dahl et al., 2013), and the ability to scale the learning process massively by parallelization on graphics processing units (GPUs). Today, deep learning is rapidly spreading in many data-rich scientific disciplines, and complements researchers' toolboxes with efficient predictive models, including the field of geosciences (Reichstein et al., 2019).

But while expectations in atmospheric sciences are high (see e.g., Dueben and Bauer, 2018; Gentine et al., 2018), the investigation of deep learning in radar-based precipitation nowcasting is still in its infancy, and universal solutions are not yet available. Shi et al. (2015) were the first to introduce deep learning models in the field of radar-based precipitation nowcasting: they presented a Convolutional Long Short-Term Memory architecture (ConvLSTM) which outperformed the optical flow





based ROVER nowcasting system in the Hong Kong area. A follow-up study (Shi et al., 2017) introduced new deep learning

architectures, namely the Trajectory Gated Recurrent Unit (TrajGRU) and the Convolutional Gated Recurrent Unit (ConvGRU), and demonstrated that these models outperform the ROVER nowcasting system, too. Further studies of Singh et al. (2017) and Shi et al. (2018) confirmed the potential of deep-learning models for radar-based precipitation nowcasting for different sites in the US and China. Most recently, Agrawal et al. (2019) introduced a U-net based deep learning model for the prediction of exceedance of specific rainfall intensity thresholds compared to optical flow and numerical weather prediction models. Hence,

the exploration of deep learning techniques in radar-based nowcasting has begun, and the potential to overcome the limitations of standard tracking and extrapolation techniques has become apparent. There is a strong need, though, to further investigate different architectures, to set up new benchmark experiments, and to understand under which conditions deep learning models can be a viable option for operational services.

In this paper, we introduce RainNet – a deep neural network which aims at learning representations of spatiotemporal

precipitation field movement and evolution from a massive open radar data archive to provide skillful precipitation nowcasts. The present study outlines RainNet's architecture and its training, and reports on a set of benchmark experiments in which RainNet competes against a conventional nowcasting model based on optical flow. Based on these experiments, we evaluate the potential of RainNet for nowcasting, but also its limitations in comparison to conventional radar-based nowcasting techniques. Based on this evaluation, we attempt to highlight options for future research towards the application of deep learning in the

field of precipitation nowcasting.

## 2 Model description

### 2.1 Network architecture

To investigate the potential of deep neural networks for radar-based precipitation nowcasting, we developed RainNet – a convolutional deep neural network (Fig. 1). Its architecture was inspired by the U-Net and SegNet families of deep learning

models for binary segmentation (Badrinarayanan et al., 2017; Ronneberger et al., 2015; Iglovikov and Shvets, 2018). These models follow an encoder-decoder architecture in which the encoder progressively downscales the spatial resolution using pooling, followed by convolutional layers; and the decoder progressively upscales the learned patterns to a higher spatial resolution using upsampling, followed by convolutional layers. There are skip connections (Srivastava et al., 2015) from the encoder to the decoder in order to ensure semantic connectivity between features on different layers.

As elementary building blocks, RainNet has 20 convolutional, four max pooling, four upsampling, two dropout layers, and four skip connections. Convolutional layers aim to generate data-driven spatial features from the corresponding input volume using several convolutional filters. Each filter is a 3D tensor of learnable weights with a small spatial kernel size (e.g., 3×3, and the third dimension equal to that of the input volume). A filter convolves through the input volume with a step size parameter (or stride, stride=1 in this study) and produces a dot product between filter weights and corresponding input volume values. A

bias parameter is added to this dot product, and the results are transformed using an adequate activation function. The purpose of the activation function is to add nonlinearities to the convolutional layer output – to enrich it to learn non-linear features.

**Figure 1.** Illustration of the RainNet architecture. RainNet is a convolutional deep neural network which follows a standard encoder-decoder structure with skip connections between its branches. See main text for further explanation.

To increase the efficiency of convolutional layers, it is necessary to optimize their hyperparameters (such as number of filters, kernel size, and type of activation function). This has been done in a heuristic tuning procedure (not shown). As a result, we use convolutional layers with up to 1024 filters, kernel sizes of $1 \times 1$ and $3 \times 3$, and linear or Rectified Linear Unit (ReLU; Nair and Hinton, 2010) activation functions.





Using a max pooling layer has two primary reasons: it achieves an invariance to scale transformations of detected features, and increases the network's robustness to noise and clutter (Boureau et al., 2010). The filter of a max pooling layer slides over the input volume independently for every feature map with some step parameter (or stride) and resizes it spatially using the *maximum* (max) operator. In our study, each max pooling layer filter is of size 2×2, applied with a stride of 2. Thus, we take

the maximum of 4 numbers in the filter region (2×2) which downsamples our input volume by factor 2. In contrast to a max pooling layer, an upsampling layer is designed for spatial upsampling of the input volume (Long et al., 2015). An upsampling layer operator slides over the input volume and fills (copies) each input value to a region that is defined by the upsampling kernel size (2×2 in this study).

Skip connections were proposed by Srivastava et al. (2015) in order to avoid the problem of vanishing gradients for the

training of very deep neural networks. Today, skip connections are a standard group of methods for any form of information transfer between different layers in a neural network (Gu et al., 2018). They allow for the most common patterns learned on the bottom layers to be reused by the top layers in order to maintain a connection between different data representations along the whole network. Skip connections turned out to be crucial for deep neural network efficiency in recent studies (Iglovikov and Shvets, 2018). For RainNet, we use skip connections for the transition of learned patterns from the encoder to the decoder

branch at the different resolutional levels.

One of the prerequisites for U-Net based architectures is that the spatial extent of input data has to be a multiple of $2^{n+1}$, where $n$ is the number of max pooling layers. As a consequence, the spatial extent on different resolutional levels becomes identical for the decoder and encoder branches. Correspondingly, the radar composite grids were transformed from the native spatial extent of 900×900 cells to the extent of 928×928 cells using mirror padding.

RainNet takes four consecutive radar composite grids as separate input channels (*t-15*, *t-10*, *t-5* minutes, and *t*, where *t* is the time of the nowcast) to produce a nowcast at time *t+5* minutes. Each grid contains 928×928 cells with an edge length of 1 km; for each cell, the input value is the logarithmic precipitation depth as retrieved from the radar-based precipitation product. There are five almost symmetrical resolutional levels for both decoder and encoder which utilize precipitation patterns at the full spatial input resolution of (*x, y*), at a quarter resolution (*x/2, y/2*), at (*x/4, y/4*), (*x/8, y/8*), and (*x/16, y/16*) respectively. To

increase the robustness and to prevent overfitting of pattern representations at coarse resolutions, we implemented a dropout regularization technique (Srivastava et al., 2014). Finally, the output layer of resolution (*x, y*) with a linear activation function provides the predicted logarithmic precipitation (in mm) in each grid cell for *t+5* minutes.

RainNet differs fundamentally from ConvLSTM (Shi et al., 2015), a prior neural-network approach, which accounts for both spatial and temporal structures in radar data by using stacked convolutional as well as LSTM layers that preserve the spatial

resolution of the input data alongside all the computational layers. LSTM networks have been observed to be brittle; in several application domains, convolutional neural networks have turned out to be numerically more stable during training, and make more accurate predictions than these recurrent neural networks (e.g., Bai et al., 2018; Gehring et al., 2017).

Therefore, RainNet uses a fully convolutional architecture, and does not use LSTM layers to propagate information through time. In order to make predictions with a larger lead time, we apply RainNet recursively. After predicting the estimated log-





precipitation for *t+5* minutes, the measured values for *t-10*, *t-5*, and *t* as well as the estimated value for *t+5* serve as the next

input volume which yields the estimated log-precipitation for *t+10* minutes. The input window is then moved on incrementally.

## 2.2   Optimization procedure

In total, RainNet has almost 31.4 million parameters. We optimized these parameters using a procedure of which we show one

iteration in Fig. 2: first, we read a sample of input data that consists of radar composite grids at time *t-15*, *t-10*, *t-5* minutes,

and *t*, and a sample of the observed precipitation at time *t+5*. For both, input and observation, we increase the spatial extent to

928×928 using mirror padding, and transform precipitation depth $x$ (in mm / 5 minutes) as follows (Eq. 1):

$$x_{transformed} = \ln(x_{raw} + 0.01) \tag{1}$$

Second, RainNet carries out a prediction based on the input data. Third, we calculate a loss function that represents the

deviation between prediction and observation. Previously, Chen et al. (2018) showed that using the *logcosh* loss function

is beneficial for the optimization of variational auto-encoders (VAE) in comparison to mean squared error. Accordingly, we

employed the *logcosh* loss function as follows (Eq. 2):

$$Loss = \frac{\sum_{i=1}^{n} \ln(\cosh(now_i - obs_i))}{n} \tag{2}$$

$$\cosh(x) = \frac{1}{2}(e^x + e^{-x}) \tag{3}$$

where $now_i$ and $obs_i$ are nowcast and observation at the $i$-th location, respectively; $\cosh$ is the hyperbolic cosine function

(Eq. 3); $n$ is the number of cells in the radar composite grid.

Fourth, we update RainNet's model parameters to minimize the loss function using backpropagation algorithm where the

Adam optimizer is utilized to compute the gradients (Kingma and Ba, 2015).

We optimized RainNet's parameters using 10 epochs (one epoch ends when the neural network saw every input data sample

once, then the next epoch begins) with a mini batch of size 2 (one mini batch holds a few input data samples). The optimization

procedure has converged on the $8^{th}$ epoch showing saturation of RainNet's performance on the validation data. The learning

rate of the Adam optimizer had a value of 1e-04, while other parameters had default values from the original paper of Kingma

and Ba (2015).

The entire setup was empirically identified as the most successful in terms of RainNet performance on validation data, while

other configurations with different loss functions (e.g., mean absolute error, mean squared error) and optimization algorithms

(e.g., stochastic gradient descent) have also converged. The average training time on a single GPU (NVIDIA GTX GeForce

1080Ti, NVIDIA GTX TITAN X, or NVIDIA Tesla P100) varies from 72 to 76 hours.



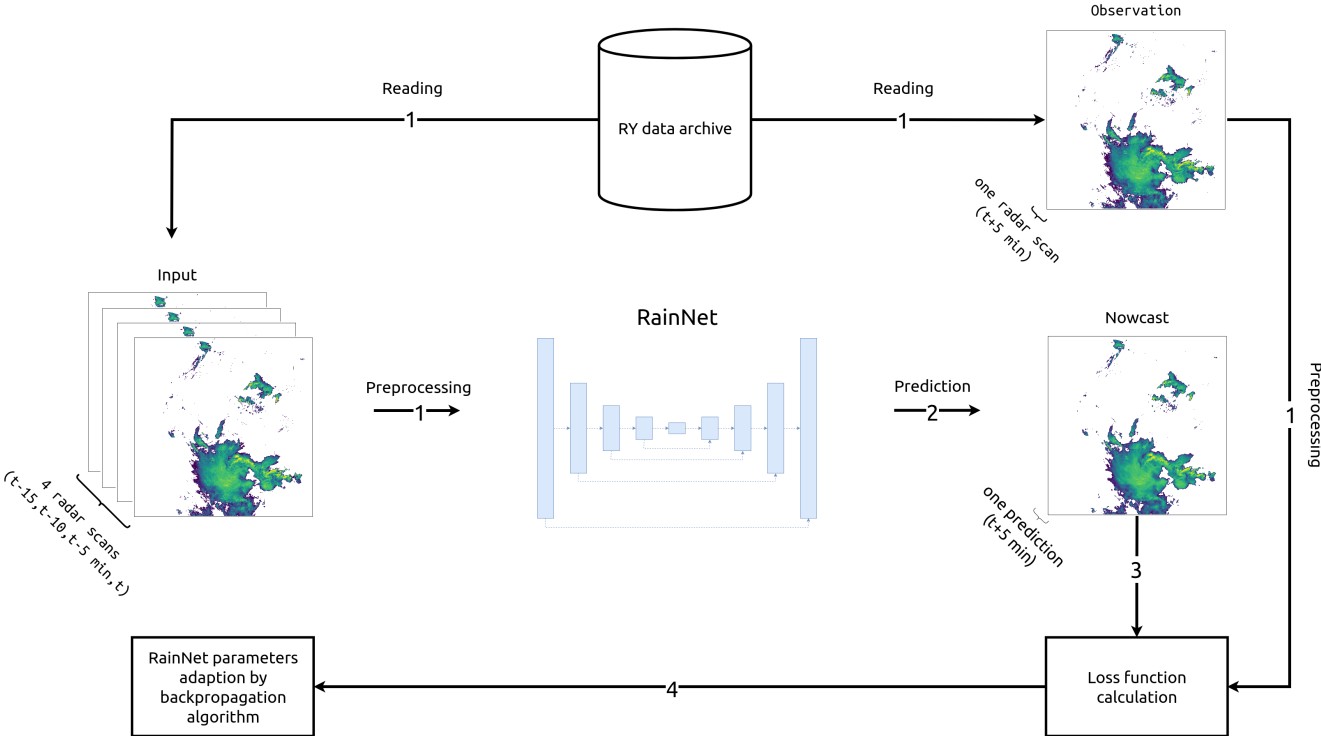

**Figure 2.** Illustration of one iteration step of the RainNet parameters optimization procedure.

We support this paper by a corresponding repository on GitHub (https://github.com/hydrogo/rainnet; Ayzel, 2020a) which holds the RainNet model architecture written in Python 3 programming language (https://python.org, last access: 28 January 2020) using the *Keras* deep learning library (Chollet et al., 2015) alongside its parameters (Ayzel, 2020b) which had been

optimized on the radar data set which is described in the following section.

## 3   Data and experimental setup

### 3.1   Radar data

We use the RY product of the German Weather Service (DWD) as input data for training and validating the RainNet model. The RY product represents a quality-controlled rainfall-depth composite of 17 operational DWD Doppler radars. It has a spatial

extent of 900×900 km, covers the whole area of Germany, and is available since 2006. The spatial and temporal resolution of the RY product is 1×1 km and 5 minutes, respectively.

In this study, we use RY data that covers the period from 2006 to 2017. We split the available RY data as follows: while we use data from 2006 to 2013 to optimize RainNet's model parameters and data from 2014 to 2015 to validate RainNet performance, data from 2016 to 2017 is used for model verification (Sect. 3.3). For both optimization and validation periods,





we keep only data from May to September and ignore time steps for which the precipitation field (with rainfall intensity more than 0.125 mm h$^{-1}$) covers less than 10% of the RY domain.

## 3.2 Reference models

We use a nowcasting models from the *rainymotion* Python library (Ayzel et al., 2019) as benchmarks against which we evaluate RainNet. As the first baseline model, we use Eulerian persistence (further referred to as Persistence), which assumes that for any lead time *n* (minutes), precipitation at *t+n* is the same as at forecast time *t*. Despite its simplicity, it is quite a powerful model for very short lead times, which also establishes a solid verification efficiency baseline which can be achieved with a trivial model without any explicit assumptions. As the second baseline model, we use the Dense model from the *rainymotion* library (further referred to as Rainymotion) which is based on optical flow techniques for precipitation field tracking and the constant-vector advection scheme for precipitation field extrapolation. Ayzel et al. (2019) showed that this model has an equivalent or even superior performance in comparison to the operational RADVOR model from DWD for a wide range of rainfall events.

## 3.3 Verification experiments and performance evaluation

For benchmarking RainNet's predictive skill in comparison to the baseline models, Rainymotion and Persistence, we selected 11 events during the summer months of the verification period (2016–2017). These events are selected for covering a range of event characteristics with different rainfall intensity, spatial coverage, and duration. A detailed account of the events' properties was given by Ayzel et al. (2019).

We use three metrics for model verification: mean absolute error (MAE), critical success index (CSI), and fractions skill score (FSS). Each metric represents a different category of scores. MAE (Eq. 4) corresponds to the continuous category and maps the differences between nowcast and observed rainfall intensities; CSI (Eq. 5) is a categorical score which is based on a standard contingency table for calculating matches between Boolean variables which indicate the exceedance of specific rainfall intensity thresholds; FSS (Eq. 6) represents neighborhood verification scores and is based on comparing nowcast and observed fractional coverage of rainfall intensities exceeding specific thresholds in spatial neighborhoods (windows) of certain size.

$$MAE = \frac{\sum_{i=1}^{n} |now_i - obs_i|}{n} \quad (4)$$

$$CSI = \frac{hits}{hits + false\ alarms + misses} \quad (5)$$

$$FSS = 1 - \frac{\sum_{i=1}^{n} (P_n - P_o)^2}{\sum_{i=1}^{n} P_n^2 + \sum_{i=1}^{n} P_o^2} \quad (6)$$





where quantities $now_i$ and $obs_i$ are nowcast and observed rainfall rate in the $i$-th pixel of the corresponding radar image and $n$ is the number of pixels. *Hits*, *false alarms*, and *misses* are defined by the contingency table and the corresponding threshold value. Quantities $P_n$ and $P_o$ represent the nowcast and observed fractions of rainfall intensities exceeding a specific threshold

for a defined neighborhood size, respectively. MAE is positive and unbounded with a perfect score of 0; both CSI and FSS can vary from 0 to 1 with a perfect score of 1. We have applied threshold rain rates of 0.125, 1, and 5 mm h$^{-1}$ for calculating CSI and FSS. For calculating FSS we use neighborhood (window) sizes of 1, 5, 10, and 20 km.

The verification metrics we use in this study quantify the models' performance from different perspectives. The MAE captures errors in rainfall rate prediction (the less the better), CSI (the higher the better) captures model accuracy – the fraction

of the forecast event that was correctly predicted – but it does not distinguish between the sources of errors. The FSS determines how the nowcast skill depends on both threshold of rainfall exceedance and spatial scale (Mittermaier and Roberts, 2010).

In addition to standard verification metrics described above, we calculate the power spectral density (PSD) of nowcasts and corresponding observations using Welch's method (Welch, 1967) to investigate the effects of smoothing demonstrated by different models.

## 210   4   Results and discussion

For each event, RainNet was used to compute nowcasts at lead times from 5 to 60 minutes (in 5-minute steps). To predict the precipitation at time *t+5* minutes (*t* being forecast time), we used the four latest radar images (at time *t-15*, *t-10*, *t-5* minutes, and *t*) as input. And since RainNet was only trained to predict precipitation at five minutes lead time, predictions beyond *t+5* were made recursively: in order to predict precipitation at *t+10*, we considered the prediction at *t+5* as the latest observation. That

recursive procedure was repeated up to a maximum lead time of 60 minutes. Rainymotion uses the two latest radar composite grids (*t-5*, *t*) in order to retrieve a velocity field, and then to advect the latest radar-based precipitation observation at forecast time *t* to *t+5*, *t+10*, ..., and *t+60*.

Fig. 3 shows the routine verification metrics MAE and CSI for RainNet, Rainymotion, and Persistence, as a function of lead time. Preliminary analysis had shown the same general pattern of model efficiency for each of the eleven events (Sect.

S1 in the Supplement), which is why we only show the average metrics over all events. Clearly, RainNet outperforms the benchmarks in each metric and lead time (differences between models were tested to be significant with the two-tailored T-test at a significance level of 5%, results not shown). Persistence is the least skillful, as could be expected for a trivial baseline. The relative differences between RainNet and Rainymotion are more pronounced for the MAE than for the CSI. For the MAE, the difference between RainNet and Rainymotion increases with lead time. For the CSI, the difference between RainNet and

Rainymotion appears to be highest for intermediate lead times between 20 and 40 minutes. The performance of all models, in terms of CSI, decreases with increasing intensity thresholds. The relative difference between RainNet and Rainymotion metrics is the lowest for the CSI at 5 mm h$^{-1}$.





**Figure 3.** Mean Absolute Error (MAE) and Critical Success Index (CSI) for three different intensity thresholds (0.125 mm h$^{-1}$, 1 mm h$^{-1}$, 5 mm h$^{-1}$). The metrics are shown as a function of lead time. All values represent the average of the corresponding metric over all 11 verification events.

In summary, Fig. 3 suggests that RainNet outperforms Rainymotion (as a representative of standard tracking and extrapolation techniques based on optical flow) for any of the metrics shown. Yet, the CSI for a threshold of 5 mm h$^{-1}$ already implies

that RainNet might have difficulties in predicting pronounced precipitation features with high intensities.

In order to better understand the fundamental properties of RainNet predictions in contrast to Rainymotion, we continue by inspecting a nowcast at three different lead times (5, 30 and 60 minutes), for a verification event at an arbitrarily selected forecast time (2016-05-29 19:15:00 UTC). The top row of Fig. 4 shows the observed precipitation, the second and third row show Rainymotion and RainNet predictions. And since it is visually challenging to track the motion pattern at the scale of

900×900 km by eyeball, we illustrate the velocity field as obtained from optical flow which forms the basis for Rainymotion's prediction. While it is certainly difficult to infer the predictive performance of the two models from this figure, another feature becomes immediately striking: RainNet introduces a spatial smoothing which appears to substantially increase with lead time.





In order to quantify that visual impression, we calculated, for the same example, the power spectral density (PSD) of the nowcasts and the corresponding observations (bottom row in Fig. 4), using Welch's method (Welch, 1967). In simple terms,

the PSD represents the prominence of precipitation features at different spatial scales, expressed as the spectral power at different wavelengths after a two-dimensional Fast Fourier Transform. The power spectrum itself is not of specific interest here; it is the loss of power at different length scales, relative to the observation, that is relevant in this context. The loss of power of Rainymotion nowcasts appears to be constrained to spatial scales below 4 km, and does not seem to depend on lead time (see also Ayzel et al., 2019). For RainNet, however, a substantial loss of power at length scales below 16 km becomes

apparent at a lead time of 5 minutes. For longer lead times of 30 and 60 minutes, that loss of power grows and propagates to scales of up to 32 km. That loss of power over a range of scales corresponds to our visual impression of spatial smoothing.

In order to investigate whether that loss of spectral power at smaller scales is a general property of RainNet predictions, we computed the PSD for each forecast time in each verification event, in order to obtain an average PSD for observations and nowcasts, at lead times of 5, 30, and 60 minutes. The corresponding results are shown in Fig. 5. They confirm that the

behaviour observed in the bottom row of Fig. 4 is, in fact, representative for the entirety of verification events. Precipitation fields predicted by RainNet are much smoother than both the observed fields and the Rainymotion nowcasts. At a lead time of five minutes, RainNet starts to lose power at a scale of 16 km. That loss accumulates over lead time and becomes effective up to a scale of 32 km at a lead time of 60 minutes. These results confirm qualitative findings of Shi et al. (2015) and Shi et al. (2018) who described their nowcasts as "smooth" or "fuzzy".

RainNet obviously learned, as the optimal way to minimize the loss function, to introduce a certain level of smoothing for the prediction at time *t+5* minutes. According to the loss of spectral power, the smoothing is still small at a length scale of 16 km, but becomes increasingly effective at smaller scales from 8 to 2 km. It is important to note that the loss of power below length scales of 16 km at a lead time of five minutes is an essential property of RainNet. It reflects the learning outcome, and illustrates how RainNet factors in predictive uncertainty at five minutes lead time by smoothing over small spatial scales. Conversely,

the increasing loss of power and its propagation to larger scales up to 32 km are *not* an inherent property of RainNet, but a consequence of its recursive application in our study context: as the predictions at short lead times serve as model input for predictions at longer lead times, the results become increasingly smooth. So while the smoothing introduced at five minutes lead time can be interpreted as a direct result of the learning procedure, the cumulative smoothing at longer lead time has to be considered rather an artifact, similar to the effect of "numerical diffusion" in numerically solving the advection equation.

Given this understanding of RainNet's properties, we used the Fractions Skill Score (FSS) to provide further insight into the dependency of predictive skill on the spatial scale. To that end, the FSS was obtained by comparing the predicted and observed fractional coverage of pixels (inside a spatial window / neighbourhood) that exceed a certain intensity threshold (see Eq. 6 in Sect. 3.3). Fig. 6 shows the FSS for Rainymotion and RainNet, as an average over all verification events, for spatial window sizes of 1, 5, 10, and 20 km, and for intensity thresholds of 0.125, 1, and 5 mm h$^{-1}$. In addition to the color code, the value

of the FSS is given for each combination of window size (scale) and intensity. In case one model is superior to the other, the correspondingly higher FSS value is highlighted in bold black digits.







**Figure 4.** Precipitation observations as well as Rainymotion and RainNet nowcasts at *t*=2016-05-29 19:15; *top row*: observed precipitation intensity at time *t*, *t+5*, *t+30* and *t+60* minutes; *second row*: corresponding Rainymotion prediction, together with the underlying velocity field obtained from optical flow; *bottom row*: power spectral density plots for observations and nowcasts at lead times 5, 30 and 60 minutes.

Based on the above results and discussion of RainNet's versus Rainymotion's predictive properties, the FSS figures are plausible, and provide a more formalized approach to express the effects of smoothing in terms of predictive skill. For a window size of 1 km – i.e., the native grid resolution – RainNet outperforms Rainymotion for each intensity threshold and lead time. That finding is consistent with the CSI as shown in Fig. 3. Yet, the superiority of RainNet successively becomes lost –





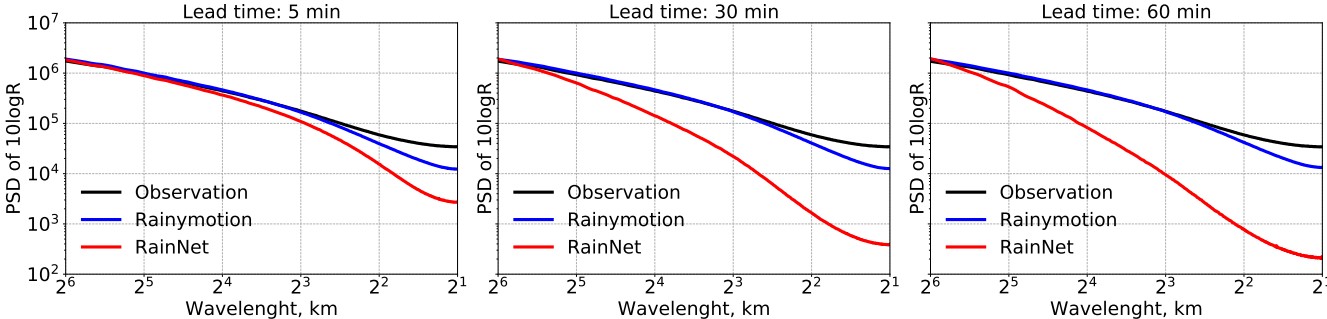

**Figure 5.** PSD averaged over all verification events and nowcasts, for lead times of 5, 30, and 60 minutes.

and even reversed – with increasing window sizes, intensity thresholds, and lead times. That effect becomes most pronounced at a window size of 20 km, an intensity of 5 mm h$^{-1}$, and a lead time of 60 minutes, where Rainymotion outperforms RainNet by an FSS of 0.64 versus 0.54, the largest difference found in the entire set of FSS values in Fig. 6. Yet, Rainymotion already starts to slightly outperform RainNet at a window size of 5 km and an intensity threshold of 5 mm h$^{-1}$ (all lead times), or a window size of 10 km and a lead time of 60 minutes (all intensity thresholds).

The dependency of the FSS (or the difference of FSS values between Rainymotion and RainNet) on spatial scale, intensity threshold, and lead time, is a direct result of inherent model properties. Rainymotion advects precipitation features, but preserves their intensity, while RainNet has not only learned how precipitation fields move in space, but that smoothing in space is an efficient way to minimize the loss function. When we increase the size of the spatial neighbourhood around a pixel, this neighbourhood could, at some size, include high intensity precipitation features that Rainymotion has preserved, but just slightly misplaced. RainNet, however, could have lost such features entirely as a result of smoothing. As discussed above, that effect becomes increasingly prominent for longer lead times because the effect of smoothing propagates.

## 5 Summary and conclusions

In this study, we have presented RainNet, a deep convolutional neural network architecture for radar-based precipitation nowcasting. Its design was inspired by the U-Net and SegNet families of deep learning models for binary segmentation, and follows an encoder-decoder architecture in which the encoder progressively downscales the spatial resolution using pooling, followed by convolutional layers; and the decoder progressively upscales the learned patterns to a higher spatial resolution using upsampling, followed by convolutional layers.

RainNet was trained to predict precipitation at a lead time of five minutes, using several years of quality-controlled weather radar composites based on the DWD weather radar network. That data covers Germany with a spatial domain of 900×900 km, and has a resolution of 1 km in space and 5 minutes in time. Independent verification experiments were carried out on eleven summer precipitation events from 2016 to 2017. In order to achieve a lead time of 60 minutes, a recursive approach was implemented by using RainNet predictions at five minutes lead time as model input for longer lead times. In the verification



Figure 6. Fractions Skill Score (FSS) for Rainymotion (*top panel*) and RainNet (*bottom panel*), for 5, 30, and 60 minutes lead time, and spatial window sizes of 1, 5, 10 and 20 km, and for intensity thresholds of 0.125, 1, and 5 mm h$^{-1}$. In addition to the color code of the FSS, we added the numerical FSS values. The FSS value of the model which is significantly superior for a specific combination of window size, intensity threshold, and lead time is typed in bold black digits, for the inferior model in regular.

experiments, Eurlerian persistence served as a trivial benchmark. As an additional benchmark, we used a model from the

*rainymotion* library which had previously been shown to outperform the operational nowcasting model of the German Weather Service for the same set of verification events.





RainNet significantly outperformed both benchmark models at all lead times up to 60 minutes for the routine verification metrics Mean Absolute Error (MAE) and Critical Success Index (CSI, at intensity thresholds of 0.125, 1, and 5 mm h$^{-1}$). Depending on the verification metric, the results correspond to an extension of effective lead time in the order of 10–20
minutes by RainNet as compared to Rainymotion. Since both Rainymotion and RainNet substantially outperformed the trivial benchmark of Persistence, the latter was not considered in subsequent analyses.

Apart from its superiority in terms of MAE and CSI, a remarkable property of RainNet predictions, in comparison to Rainymotion, is the level of spatial smoothing. That smoothing becomes increasingly apparent at longer lead times. Yet, it is already prominent at a lead time of five minutes. That was confirmed by an analysis of Power Spectral Density which
showed, at time *t+5* minutes, a loss of spectral power at length scales of 16 km and below. Obviously, RainNet has learned an optimal level of smoothing to produce a nowcast at 5 minutes lead time. In that sense, the loss of spectral power at small scales is informative, as it reflects the limits of predictability as a function of spatial scale. Beyond the lead time of five minutes, however, the increasing level of smoothing is a mere artifact – an analogue to numerical diffusion – that is not a property of RainNet itself, but of its recursive application: as we repeatedly use smoothed nowcasts as model inputs, we cumulate the effect
of smoothing over time. That certainly is an undesirable property, and it becomes particularly unfavourable for the prediction of high-intensity precipitation features. As was shown on the basis of the Fractions Skill Score (FSS), Rainymotion outperforms RainNet for intensive precipitation (>5 mm h$^{-1}$) once we start to evaluate performance in a spatial neighbourhood around the native grid pixel of 1×1 km size. This is because Rainymotion preserves distinct precipitation features, but tends to misplace them. RainNet, however, tends to lose such features over longer lead times due to cumulative smoothing effects – more so if it
is applied recursively.

From an early warning perspective, that property of RainNet limits its usefulness. There are, however, options to address that issue in future research:

– The loss function used in the training could be adjusted in order to penalize the loss of power at small spatial scales. The loss function explicitly represents our requirements to the model. Verifying the model by other performance metrics
325        will typically reveal whether these metrics are rather in agreement or in conflict with these requirements. In our case, the *logcosh* loss function appears to favour a low MAE, but at the cost of losing distinct precipitation features. In general, future users need to be aware that, apart from the network design, the optimization itself constitutes the main difference to "heuristic" tracking-and-extrapolation techniques (such as Rainymotion) which do not use any systematic parameter optimization. The training procedure will stubbornly attempt to minimize the loss function, irrespective of
330        what researchers consider as "physically plausible". For many researchers in the field of nowcasting, that notion might be in stark contrast to experiences with "conventional" nowcasting techniques which tend to effortlessly produce at least plausible patterns;

– RainNet should be directly trained to predict precipitation at lead times beyond five minutes. However, preliminary training experiments with that learning task had difficulties to converge. We thus recommend to still use recursive predictions
335        as *model input* for longer lead times during training, in order to improve convergence. For example, to predict precipita-





tion at time *t+10* minutes, RainNet could be trained using precipitation at time *t-15, t-10, ..., t* minutes as input, but using the recursive prediction at time *t+5* as an *additional* input layer, too. But while the direct prediction of precipitation at longer lead times should reduce excessive smoothing as a result of numerical diffusion, we would still expect the level of smoothing to increase with lead time, as a result of the predictive uncertainty at small scales;

– As an alternative to predict continuous values of precipitation intensity, RainNet could be trained to predict the exceedance of specific intensity thresholds instead. That would correspond to a binary segmentation task. It is possible that the objective of learning the segmentation for *low* intensities might be in conflict with learning it for *high* intensities. That is why the training could be carried out both separately and jointly for disparate thresholds, in order to investigate whether there are inherent trade-offs. From an early warning perspective, it makes sense to train RainNet for binary

segmentation based on user defined thresholds that are governed by the context of risk management. The additional advantage of training RainNet to predict threshold exceedance is that we could use its output directly as a measure of uncertainty (of that exceedance).

We consider any of those options worth being pursued in order to increase the usefulness of RainNet in an early warning context – i.e. to better represent precipitation intensities that exceed hazardous thresholds –, and we would expect the overall

architecture of RainNet to be a helpful starting point.

Yet, the key issue of precipitation prediction – the anticipation of convective initialization as well as the growth and dissipation of precipitation in the imminent future – still appears to be unresolved. It is an inherent limitation of nowcasting models purely based on optical flow: they can extrapolate motion fairly well, but they cannot predict intensity dynamics. Deep learning architectures, however, *might* be able to learn recurrent patterns of growth and dissipation, although it will be challenging to

verify if they actually *did*. In the context of this study, though, we have to assume that RainNet has rather learned the representation of motion patterns instead of rainfall intensity dynamics: for a lead time of five minutes, the effects of motion can generally be expected to dominate over the effects of intensity dynamics, which will propagate to the learning results. The fact that we actually could recursively use the RainNet predictions at five minutes lead time in order to predict precipitation at one hour lead time also implies that RainNet, in essence, learned to represent motion patterns and optimal smoothing.

Another limitation in successfully learning patterns of intensity growth and dissipation might be the input data itself. While we do not exclude the possibility that such patterns could be learned from just two-dimensional radar composites, other input variables might add essential information on imminent atmospheric dynamics – the predisposition of the atmosphere to produce or to dissolve precipitation. Such additional data might include 3-dimensional radar volume data, dual-pol radar moments, or output fields of numerical weather prediction (NWP) models. Formally, the inclusion of NWP fields in a learning framework

could be considered as a different way of assimilation, combining – in a data-driven way – the information content of physical models and observations.

Our study provides, after Shi et al. (2015, 2017, 2018), another proof-of-concept that convolutional neural networks provide a firm basis to compete with conventional nowcasting models based on optical flow (most recently, Google Research has also reported similar attempts based on a U-Net architecture, see Agrawal et al. (2019)). Yet, this study should rather be

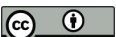



considered as a starting point: to further improve the predictive skill of convolutional neural networks, and to better understand
the properties of their predictions – in a statistical sense, but also in how processes of motion and intensity dynamics are
reflected. To that end, computational complexity and the cost of the training process still have to be considered as inhibitive,
despite the tremendous progress achieved in the past years: RainNet's training would require almost a year on a standard
desktop CPU, in contrast to the three days on a modern desktop GPU (although the latter is a challenge to implement for non-

experts). Yet, it is possible to run deep learning models with already optimized (pretrained) weights on a desktop computer.
Thus, it is important not only to make available the code of the network architecture, but also the corresponding weights,
applicable using open source software tools and libraries. We provide all this – code, pretrained weights, as well as training
and verification data – as an input to future studies on open repositories (Ayzel, 2020a, b, c).

*Code and data availability.* The RainNet model is free and open source. It is distributed under the MIT software license which allows

unrestricted use. The source code is provided through a GitHub repository https://github.com/hydrogo/rainnet (last access: 30 January 2020);
a snapshot of Rainnet v1.0 is also available at http://doi.org/10.5281/zenodo.3631038 (Ayzel, 2020a); the pretrained RainNet model and
its weights are available at http://doi.org/10.5281/zenodo.3630429 (Ayzel, 2020b). DWD provided the sample data of the RY product; it is
available at http://doi.org/10.5281/zenodo.3629951 (Ayzel, 2020c).

*Author contributions.* GA developed the RainNet model, carried out the benchmark experiments, and wrote the manuscript; TS and MH

supervised the study and co-authored the manuscript.

*Competing interests.* The authors declare that they have no conflict of interest.

*Acknowledgements.* GA was financially supported by Geo.X, the Research Network for Geosciences in Berlin and Potsdam (Project-number:
SO_087_GeoX). GA would like to thank Open Data Science community (ods.ai) for many valuable discussions and educational help in the
growing field of deep learning. We ran our experiments using GPU computation resources of the Machine Learning Group of the University

of Potsdam (Potsdam, Germany) and the Shared Facility Center "Data Center of FEB RAS" (Khabarovsk, Russia). We acknowledge the
support of Deutsche Forschungsgemeinschaft (German Research Foundation) and the Open Access Publication Fund of Potsdam University.



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
