# Peer review of "RainNet v1.0: a convolutional neural network for radar-based precipitation nowcasting"

_Geoscientific Model Development, 2020_

## Referee Comment (RC1) · Gabriele Franch (Referee) · 13 Apr 2020

**General comments**

The paper presents a deep-learning based model for the nowcasting of radar-based precipitation fields. While the novelty from the architectural point of the deep learning modeling is limited (an adaptation of the U-Net architecture), the paper makes a very good case for the application of a tried-and-true deep learning architecture to the domain of radar-based precipitation nowcasting. Overall, the presented work is very well written: the relevant concepts and references are introduced, the evaluation framework along with experimental setup and analysis is presented with clarity, and the model results are compared with proper baselines using both continuous (MAE) and

categorical scores (CSI, FSS). The discussion addresses the main challenge posed by deep learning nowcasting models: the smoothing of the predicted precipitation field over time. Last but not least, the supplementary material provides the model code, the data, the trained weights and a ready to use Colab notebook for reproducibility. Therefore, I recommend this study for publication after considering the minor comments listed below.

**Specific comments**

1. L: 167-172 can you provide the exact number of the train (optimization), validation and test (verification) sequences? Moreover, can you explain how are the sequences extracted from the dataset (are the sequences extracted using an overlapping rolling window over the selected time periods?)

2. Given the analysis of the power spectrum and the reported smoothing in the prediction, it seems that RainNet may suffer from a severe underestimation of high rainfall rates. In this regard, it would be extremely beneficial to include a higher rain rate than 5mm/h for the analysis of the categorical scores. This would also help provide a better comparison with Rainymotion and can help to answer Line 230 that states: "RainNet might have difficulties in predicting pronounced precipitation features with high intensities". Thus, I suggest adding also at least one heavy rainfall threshold (FSS and CSI $\geq 15$ mm/h) in the analysis.

---

## Referee Comment (RC2) · Scott Collis (Referee) · 14 Apr 2020

This paper by Ayzel and co-authors is a very good well references and refreshing open and honest appraisal of a Convolution Neural Net approach to precipitation nowcasting.

RainyNet, which differs from traditional "Next video frame prediction" approaches which use a LSTM plus CNN is compared to an optical flow type technique and shown to have good skill but suffers from oversmoothing thus degrading its ability to predict high intensity rainfall rates.

I appreciated this honest distinction which helps inform readers and, since the software is completely open and community based, it also helps inform potential users.

The language is clear and very readable. The figures are of reasonable quality and

the author spends a good amount of time explaining the underlying Machine Learning literature making this a great "on-ramp" for anyone beginning in this field.

I only have four minor suggestions (and they are just that, suggestions): 1) The training and prediction was entirely based on DWD RY gridded rainfall product. I think the authors should discuss the applicability of training a network on one data set and application to another. The CNN learns what features propagate and dissipate (an advantage over purely advective techniques) but this may not apply in regions where different physics dominate. 2) It would be good for the Authors to discuss a little more on what would be required of the input data. Can a potential user train with any NDArray style data? 3) One line 155 where training times are discussed it would be good, for the understanding of readers to restate how many frames (radar time steps) were used in the training. This would be a repetition but I believe it would add to the readers understanding. 4) In the author's section on future research I am surprised not to see other atmospheric data inputs/layers talked about. If I understood the paper correctly the CNN is trained purely on image-like data with no environmental awareness. I wonder the evolution (again information that can not be deduced by simple advection) could be better predicted with information like precipitable water or information about terrain? The developing area of physics aware machine learning could be an area to explore.

In conclusion, the paper by Ayzel is a very nicely written description of a new and novel technique for precipitation nowcasting. It will make an excellent paper for scientists who are looking to learn more about applied machine learning to read. The software is open source community software so reproducibility and usability is a given. This paper meets and exceeds the standards for GMD and should be accepted. -Scott Collis

---

## Author Comment (AC1) · 6 May 2020

The comment was uploaded in the form of a supplement:
https://www.geosci-model-dev-discuss.net/gmd-2020-30/gmd-2020-30-AC1-supplement.pdf

---

## Author Response (AR1)

**Final response in the interactive discussion**

Dear Referees, dear Editor,

We would like to thank you very much for your positive comments and constructive suggestions to our manuscript *"RainNet v1.0: a convolutional neural network for radar-based precipitation nowcasting"*.

In this document, we would like to provide our responses to the comments of each of the referees in one single document and to outline the corresponding changes to the manuscript. We will represent the referee comments in **bold** font, and our responses in normal font. Quotations from the original manuscript will be in *italics*, changes as part of the manuscript revision will be highlighted as underlined. For the sake of clarity and brevity, we have omitted the introductory parts of the referee reports (this omittance is marked as [...]).

We hope that our response together with the revision of the manuscript sufficiently addresses the referees' concerns.

Sincerely,
Georgy Ayzel (on behalf of the author team)

**Referee comment #1 (by Gabriele Franch)**

**[...] I recommend this study for publication after considering the minor comments listed below:**

1. **In ll. 167-172, can you provide the exact number of the train (optimization), validation and test (verification) sequences? Moreover, can you explain how the sequences are extracted from the dataset (are the sequences extracted using an overlapping rolling window over the selected time periods?)**

We thank the referee for the suggestion: the required details should, in fact, be provided to the reader. We will add, in the revised manuscript, the corresponding information (i.e. the number of sequences and whether the sequences overlap) at the end of the paragraph that was mentioned by the referee - as follows (ll. 172-175 of the revised manuscript):

*In this study, we use RY data that covers the period from 2006 to 2017. We split the available RY data as follows: while we use data from 2006 to 2013 to optimize RainNet's model parameters and data from 2014 to 2015 to validate RainNet performance, data from 2016 to 2017 is used for model verification (Sect. 3.3). For both optimization and validation*

*periods, we keep only data from May to September and ignore time steps for which the precipitation field (with rainfall intensity more than 0.125 mm h−1) covers less than 10% of the RY domain. For each subset of the data - for optimization, validation, and verification -, every time step (or frame) is used once as $t_0$ (forecast time) so that the resulting sequences that are used as input to a single forecast ($t_0$-15 min, ..., $t_0$) overlap in time. The number of resulting sequences amounts to 41988 for the optimization, 5722 for the validation, and 9626 for the verification (see also Sect. 3.3).*

2. **Given the analysis of the power spectrum and the reported smoothing in the prediction, it seems that RainNet may suffer from a severe underestimation of high rainfall rates. In this regard, it would be extremely beneficial to include a higher rain rate than 5mm/h for the analysis of the categorical scores. This would also help provide a better comparison with Rainymotion and can help to answer Line 230 that states: "RainNet might have difficulties in predicting pronounced precipitation features with high intensities". Thus, I suggest adding also at least one heavy rainfall threshold (FSS and CSI≥15mm/h) in the analysis.**

We agree that a threshold of 5 mm/h does not qualify as heavy rainfall, although the results for a threshold of 5 mm/h already provide a good impression about the general effects of an increasing intensity threshold for a categorical metric such as the CSI: first, a strong loss of skill over *all* lead times and *all* competing methods, and, second, a relatively stronger loss of skill for RainNet in comparison to rainymotion.

For intensity thresholds of 10 and 15 mm/h, these effects become much more prominent, so that rainymotion, in fact, outperforms RainNet (in terms of CSI) - particularly at intermediate lead times between from 10 to 50 minutes. At the same time, the CSI of both methods becomes so low that neither can be really considered as skillful in predicting the exceedance of higher thresholds.

Hence, we would like to thank the referee very much for his suggestion. Looking at thresholds of 10 and 15 mm/h is in fact revealing: while the results basically confirm our assumptions that were based on the 5 mm/h threshold, they demonstrate more clearly how difficult it is for RainNet to learn the prediction of high intensity features. Whether this is an effect of the spatial smoothing, or whether RainNet has specifically learned, based on the current training setting and loss function, that it is efficient to "attenuate" high rainfall intensities, will be subject to future research.

As a consequence, we will include these additional results in the main manuscript: Fig. 3 will be extended by a CSI for 10 and 15 mm/h, and Fig. 6 will be extended by the FSS values that correspond to threshold intensities of 10 and 15 mm/h (please see the new versions of these figures below). The main text in Section 4 (Results and discussion) and Section 5 (Summary and conclusions) will be changed accordingly in the revised version of the manuscript in order to explicitly account for these additional results.

[Figure]

Figure 3 (revised). Mean Absolute Error (MAE) and Critical Success Index (CSI) for three different intensity thresholds (0.125 mm h⁻¹, 1 mm h⁻¹, 5 mm h⁻¹, 10 mm h⁻¹, 15 mm h⁻¹). The metrics are shown as a function of lead time. All values represent the average of the corresponding metric over all 11 verification events.

[Figure]

Figure 6 (revised). Fractions Skill Score (FSS) for Rainymotion (top panel) and RainNet (bottom panel), for 5, 30, and 60 minutes lead time, and spatial window sizes of 1, 5, 10 and 20 km, and for intensity thresholds of 0.125, 1, 5, 10 and 15 mm h$^{-1}$. In addition to the color code of the FSS, we added the numerical FSS values. The FSS value of the model which is significantly superior for a specific combination of window size, intensity threshold, and lead time is typed in bold black digits, for the inferior model in regular.

**Referee comment #2 (by Scott Collis)**

**[...] I only have four minor suggestions (and they are just that, suggestions):**

**1. The training and prediction was entirely based on DWD RY gridded rainfall product. I think the authors should discuss the applicability of training a network on one data set and application to another. The CNN learns what features propagate and dissipate (an advantage over purely advective techniques) but this may not apply in regions where different physics dominate.**

We fully agree with the referee. If the network was, in fact, successful in learning initiation, growth and dissipation of features, the transferability of such a trained network to a different region of application would most likely be limited. That limitation would be most pronounced in areas and situations in which precipitation dynamics are dominated by recurrent drivers and processes (such as features of the general circulation or orography). In ll. 351-359 of the original manuscript, however, we already pointed out that the current (trained) network is most likely very limited in its ability to represent precipitation dynamics. While we hope to change that in the future, it implies, in turn, that the transferability to another region/dataset might in fact not be as low as expected. Yet, there is only one way to find out: by actually

carrying out a verification experiment with RainNet on a dataset from another region, using the pretrained weights... another item on the to-do list for future research.

We would like to avoid to put too much emphasis on this discussion in the manuscript (as it is quite speculative), but we will account for the referee's suggestion by adding the following sentence to the paragraph in ll. 376-377 of the revised manuscript:

*[...] implies that RainNet, in essence, learned to represent motion patterns and optimal smoothing. In that case, the trained model might even be applicable on data in another region which could be tested in future verification experiments.*

**2. It would be good for the authors to discuss a little more on what would be required of the input data. Can a potential user train with any NDArray style data?**

In principle, the input data is of class numpy.ndarray (float32 with no missing values). However, the dataset used for training is so large that it cannot be read into memory at once. We store data directly in `.npy` files as it is more efficient in terms of parallelization and utilization of computational resources. While we are happy to provide this response to the referee, we would prefer not to go into these technical details in the manuscript.

**3. On line 155 where training times are discussed it would be good, for the understanding of readers to restate how many frames (radar time steps) were used in the training. This would be a repetition but I believe it would add to the readers understanding.**

We assume that the referee refers to the number of time steps or frames used as training data, not the number of frames that are used as model input for a single prediction (which is the four recent frames, see l. 116 of the revised manuscript). Stating that information, as required by the referee, would not be repetition as it has not been stated in the original manuscript. In fact, the referee's request is well in line with the first issue raised by referee #1 (please see above). We would prefer, however, to provide the required details in ll. 172-175 of the revised manuscript (not, as suggested by the referee, around l. 155 of the original manuscript).

**4. In the author's section on future research I am surprised not to see other atmospheric data inputs/layers talked about. If I understood the paper correctly the CNN is trained purely on image-like data with no environmental awareness. I wonder if the evolution (again information that can not be deduced by simple advection) could be better predicted with information like precipitable water or information about terrain? The developing area of physics aware machine learning could be an area to explore.**

We would like to refer to ll. 360-366 of the original manuscript. In that paragraph, we already outline the perspective to use output fields of atmospheric models or 3-d polarimetric radar moments as input layers that are more physically meaningful. We hope that pointing out these perspectives is sufficient to meet the referee's requirements.

[revised manuscript text omitted]